# Effects of between and within Herd Moves on Elephant Endotheliotropic Herpesvirus (EEHV) Recrudescence and Shedding in Captive Asian Elephants (*Elephas maximus*)

**DOI:** 10.3390/v14020229

**Published:** 2022-01-24

**Authors:** Sanna Eriksson Titus, Stuart Patterson, Joanna Prince-Wright, Akbar Dastjerdi, Fieke Marije Molenaar

**Affiliations:** 1Institute of Zoology, Zoological Society of London, Regents Park, London NW1 4RY, UK; 2Royal Veterinary College, Royal College Street, London NW1 0TU, UK; spatterson@rvc.ac.uk; 3Animal and Plant Health Agency (APHA)-Weybridge, Woodham Lane, New Haw, Addlestone, Surrey KT15 3NB, UK; Joanna.Prince-Wright@apha.gov.uk (J.P.-W.); akbar.dastjerdi@apha.gov.uk (A.D.); 4ZSL Whipsnade Zoo, Part of the Zoological Society of London, Regents Park, London NW1 4RY, UK; fieke.molenaar@zsl.org

**Keywords:** *Elephas maximus*, elephant endotheliotropic herpesvirus, recrudescence, management, social change, qPCR

## Abstract

Haemorrhagic disease associated with elephant endotheliotropic herpesvirus (Elephantid herpesvirus, EEHV) infections is the leading cause of death for Asian elephant (*Elephas maximus*) calves. This study assessed the effect of captive herd management on EEHV shedding, as evidence of latent infection reactivation, focusing on: (1) the influence of social change on the odds of recrudescence; (2) the respective effects of between and within herd moves; and (3) characteristics of recrudescent viral shedding. Trunk and conjunctival swabs (*n* = 165) were obtained from six elephants at an EAZA-accredited zoo, collected during a period of social stability, and at times of social change. Longitudinal sampling took place at times of moving two bulls out of the collection and one new bull into an adjacent enclosure to the cow herd (between herd moves), and during a period of mixing this new bull with the cow herd to facilitate mating (within herd moves). Quantitative PCR was employed to detect EEHV 1a/b, 4a/b, and EF–1–α (housekeeping gene). Generalised estimating equations determined EEHV recrudescence odds ratios (OR) and relative viral DNA load. Sixteen EEHV 1a/b shedding events occurred, but no EEHV 4a/b was detected. All management-derived social changes promoted recrudescence (social change OR = 3.27, 95% CI = 0.412–26, *p* = 0.262; and between herd moves OR = 1.6, 95% CI = 0.178−14.4, *p* = 0.675), though within herd movements posed the most significant increase of EEHV reactivation odds (OR = 6.86, 95% CI = 0.823−57.1, *p* = 0.075) and demonstrated the strongest relative influence (post hoc Tukey test *p* = 0.0425). Shedding onset and magnitude ranged from six to 54 days and from 3.59 to 11.09 ΔCts. Differing challenges are associated with between and within herd movements, which can promote recrudescence and should be considered an exposure risk to naïve elephants.

## 1. Introduction

The haemorrhagic disease (HD) process associated with clinical elephant endotheliotropic herpesvirus (elephantid herpesvirus, EEHV) infections is the leading cause of death in captive Asian elephant (*Elephas maximus*) calves in Europe and North America [1,2]. This virus is responsible for 20% of all Asian elephant deaths in Western zoos over the last 30 years and 65% of all fatalities in North America [2,3]. Between 1985 and 2017, EEHV-HD caused 57% of calf mortalities in Europe [1]. For the past forty years, Asian elephants have been categorised as ‘endangered’ and decreasing on the International Union for the Conservation of Nature’s Red List [4]. Captive collections are especially vulnerable to HD mortalities [1,5], which are essential for species conservation by serving as a reserve for genetic diversity, research, and future reintroduction initiatives. Both in- and ex-situ breeding programmes are presently unsustainable due to the effects of EEHV-HD, and if current trends continue, captive Asian elephants in North America will be demographically extinct before 2050 [6,7,8]. Intense monitoring and prevention remain the most efficient approach to controlling infections and the risk of HD-associated death, as EEHV lacks an effective antiviral treatment and vaccination strategy. Key considerations to understand EEHV epidemiology include prevalence, exposure routes, drivers of recrudescence, and the role of latent carriers. Within the last decade, EEHV prevalence and exposure routes have been defined, but researchers have called for greater understanding of recrudescent cases and actions promoting viral shedding [5,6,9].

Taxonomised into an exclusive *Proboscivirus* genus within the *Herpesviridae* family, there are presently seven known EEHV genotypes (EEHV-1 to -7) [1,5]. Asian elephants, the most likely species to develop clinical HD, are susceptible to 1a/b, 4a/b and 5a/b [5,10]. Polymerase chain reactions (PCR) and DNA sequencing are widely used to differentiate genotypes and have proven that each captive collection harbours unique EEHV strains [11,12,13]. Cross protection against different variants post-exposure has not been proven, co-infection is possible, and pathogenicity varies [14,15,16]. For primary infections, incubation can last up to two weeks, followed by viraemia and onset of viral shedding about a week later [17]. Although transmission routes have not yet been proven, mucosal secretion and saliva are the most probable avenues of viral exchange [17,18,19]. Between herd moves, which involves the transport of elephants between facilities, may allow restricted trunk contact, and can therefore expose the resident herd to EEHV through saliva and trunk secretions. Within herd moves is synonymous with unrestricted direct mingling, thus exposing other elephants to potentially infected bodily fluids when mating. EEHV 1a is responsible for 80% of HD deaths in European zoos (20 cases total between 1985 to 2017) and 1b appears the second-most pathogenic [1,20]. Both EEHV 4a/b and 5a/b infections have also each led to one HD fatality in Europe [15,21,22]. Salivary glands and the gastrointestinal system serve as tissue tropism for EEHV 1a and EEHV 4 infections [18]. Calves with EEHV-HD demonstrate increased infiltration of Iba-1-positive macrophages in the inflamed tissues of internal organs [23]. The nuclei of sublingual salivary glands are typically more enlarged and greater lymphohistiocytic inflammatory cell infiltration occurs in EEHV 1a cases when compared to EEHV 4 [18]. Inversely, calves with EEHV 4 sometimes exhibit higher instances of gastric mucosal haemorrhage than EEHV 1a infections [18]. Cellular apoptosis is elevated within the tissues of EEHV-HD patients, particularly within peripheral blood mononuclear cells [23]. Elephants experiencing primary exposure to high viral loads are susceptible to HD [5,24,25]. High EEHV viral loads are correlated to severe clinical signs, which, for example, may develop from lethargy and petechiae to oedema and cyanosis [26,27]. HD is particularly challenging to manage due to its rapid progression from clinical signs to per acute death, which can occur within 36 hours [28,29,30]. The mortality rate has recently improved due to rapid veterinary intervention and improved treatment before significant vascular damage occurs [31,32]. Nevertheless, 70% of primary global EEHV infections have been fatal within one to seven days [5,28,33].

Parallel to other herpesviruses, elephants who survive primary EEHV infections remain life-long carriers [5,11,30]. Recrudescence of latent EEHV infections is attributed to immunosuppression, typically involving stress or chemical agents [25,34]. Virus circulation among asymptomatic individuals has been documented [6,30,35], and a study by Hoornweg et al. (2021) suggests EEHV omnipresence, as 97.5% (*n* = 41) and 100% (*n* = 69) of screened Asian elephants from European zoos and Laotian semi-captivity were seropositive, respectively [3]. Hence, some researchers advise all adult captive Asian elephants should be considered as EEHV carriers [1,36,37]. The danger posed by latent carriers to naïve individuals, however, remains largely unexplored. It is not known at which viral threshold EEHV is infectious, how fast viral shedding commences following a trigger event, or the duration of shedding.

Some facilities, interestingly, have not experienced fatal HD, such as Kölner Zoo (Germany) and Zoo Emmen (the Netherlands) [38]. A novel meta-analysis of European Association of Zoos and Aquariums (EAZA) member institutions found that the only significant risk factor for infected calves to succumb to HD death was a history of fatal EEHV-HD at the facility [1]. It has been hypothesised that these zoos circulate less pathogenic or virulent strains [2,5], or retain minimal prevalence by means of genetic factors [1]. This article will explore whether it is also possible that particular management factors can influence viral loads.

Literature exploring the relationship between management interventions and EEHV reactivation is limited. Hengtrakul et al. (2020) found that construction, new keepers, and changing holding areas had no effect on EEHV shedding [39]. Otherwise, EEHV recrudescent cases are sporadically published, and the majority describe inter-herd circulation [33,40,41]. There are two reports documenting instances of increased EEHV shedding following social change [42,43]. The first demonstrates the effect of a between herd move, where two calves actively shedding EEHV presented inverse responses following the arrival of a new bull: the male experienced a peak of EEHV 5a/b viral loads while the female stabilised EEHV 1a/b shedding loads [42]. Bennett et al. (2015) researched whether pregnancy increased EEHV shedding, but instead documented six months of continuous shedding from the herd following a change in matriarchal hierarchy [43]. Though these examples illustrate an association between social change and viral shedding, the isolated strength and direction of this relationship have yet to be established.

This study, therefore, aims to assess the effect of specific changes in Asian elephant herd management on EEHV 1a/b and 4a/b shedding, as evidence of latent infection recrudescence. The objectives are to determine whether social change influences the odds of EEHV recrudescence or shedding load for individual elephants, analyse the isolated effect of between and within herd moves, and evaluate the characteristics of recrudescent viral shedding. Conclusions from this research may illustrate how varying degrees of management-derived social change impacts in-herd EEHV epidemiology, and the risk associated with recrudescent cases to susceptible individuals. These findings may additionally inform EEHV-HD prevention and control strategies by determining which management actions are appropriate when at-risk individuals are present.

## 2. Materials and Methods

### 2.1. Study Animals

Five adults from a herd of six Asian elephants at an EAZA-accredited UK zoo (1 breeding bull, 4 adult cows, 1 unsampled juvenile cow) provided a total of 165 samples. This herd had previously endured six EEHV viraemias in calves: four fatal HD cases (three from EEHV 1a and one from EEHV 1b) and two surviving EEHV 1a. All adults previously tested positive for EEHV 1a and 4 shedding; therefore, all research participants are considered EEHV 1a and 4 carriers. At the conclusion of swab collections, the Bull was 14, Cow 1 was 12, Cow 2 was 22, and both Cows 3 and 4 were 39 years old. Cow 3 is the dam of Cow 1. Cow 1 was pregnant eight months prior to the 2019 disease screening period, but there were complications, and the calf only survived a few days. Cow 2 was due to give birth in October 2019, but labour did not progress and the foetus was retained. The bull was born at a German zoo and was later transferred to a bachelor herd at a Belgian zoo, both of which were EAZA member institutions. Cow 2 was born at a different EAZA-accredited UK-based facility. Cows 3 and 4 originated from Myanmar. Management diaries were also consulted to identify any other potentially stressful events that could be associated with shedding, such as conspecific aggression, training or veterinary procedures.

### 2.2. Sample Collection

Samples were collected twice weekly throughout two three-month annual virus screening periods. Eight conjunctival and 157 trunk swabs were obtained as previously described adhering to EAZA protocols [31,44,45], and stored frozen (−20 °C) [39]. Samples were transported from the UK zoo to the Royal Veterinary College (Royal College Street, London, UK) for DNA extractions, and subsequently shipped overnight to the Animal and Plant Health Agency-Weybridge (APHA, Woodham Lane, New Haw, Addlestone, Surrey, UK) for PCR.

### 2.3. Viral Analysis

#### 2.3.1. DNA Extraction

DNA extractions were performed at the Royal Veterinary College with the commercial QIAGEN DNeasy™ Blood and Tissue Kit (QIAGEN Inc., Hilden, Germany). Swabs were prepared according to Dastjerdi et al. (2016) with one modification: rather than using glass beads (due to a lack of availability), an additional three-minute centrifugation at 6000× *g* occurred following acclimation to room temperature [31]. DNA extractions adhered to the protocol described by Hardman et al. (2012) [6]. A spectrophotometer (DS-11 FX+, DeNovix Inc, Wilmington, DE, USA) measured DNA concentrations and absorption profiles (260/280 nm and 260/230 nm) to ensure extraction success. Products were transferred to sterile 96-well plates, sealed and stored at −20 °C.

#### 2.3.2. Quantitative PCR (qPCR)

Quantitative PCRs were conducted at APHA-Weybridge with the QIAGEN QuantiFast Pathogen + IC Kit™ (QIAGEN Inc., Hilden, Germany) as described by Dastjerdi et al. (2016) and cycling conditions of 95 °C for 5 min followed by 43 cycles of 95 °C for 15 s and 60 °C for 30 s [31]. Each sample was extracted and tested once alongside a negative extraction control (EEHV negative DNA extract), a no-template control (140 µL water), and positive template control (PTC, HPLC purified 100 base synthetic oligonucleotides purchased from Eurofins Genomics, Wolverhampton, UK). The relevant primers, hydrolysis probes, and PTCs were used to detect EEHV 1a/b [30], 4a/b [46], and the alpha-gene of the translation elongation factor protein (EF–1–α) (Table 1). A valid assay produced a FAM fluorescence signal for the PTC and no signal for the negative and no-template controls. Cycle thresholds (Ct) of ≤43 were accepted as EEHV 1a/b, 4a/b and EF–1–α positive. Detection of the housekeeping gene EF–1–α confirms DNA extraction and qPCR performance [26,47]. If EF–1–α was not detected, results were considered inconclusive and excluded from further analysis. This study omitted qPCR analysis for EEHV 5a/b due to lack of a validated PCR.

#### 2.3.3. Data Normalisation

By employing a portion of the method described by Pfaffl (2001), various samples can be compared via normalising EEHV shedding to EF–1–α gene expression [48,49]:Normalised shedding = ΔCt = Ct _EEHV 1a/b or 4a/b_ − Ct _EF–1–α_(1)

Normalised shedding allows equal interpretation of results across facilities, experimental settings, and sample types [49]. Cts describe the number of qPCR cycles necessary to detect primer-specific DNA; therefore, samples producing higher Cts possess less target DNA and, accordingly, higher ΔCt results indicate less EEHV shedding [6]. Figures visualising normalised shedding accordingly display inverted y-axes to describe this relationship.

### 2.4. Management Periods

Management actions were categorised into periods of social change (between and within herd moves) and social stability, which served as a control (Figure 1). ‘Between herd moves’ describes relocating two bulls out of the collection, three days later importing a new bull (auditory and olfactory contact only), and 19 days later allowing barrier-protected exposure (restricted trunk contact with the herd). ‘Within herd moves’ represents mixing the novel elephant with the cow herd for breeding purposes (i.e., unrestricted direct contact). To evaluate the effects of social change on EEHV recrudescence, samples were assigned to a management period. Any sample collected after a management event could reveal residual effects, so all dates after an event were designated within the same management period. Although the period of social stability commenced 11 months after the period of between herd moves, a lack of continuous shedding in this timeframe indicated its suitability as a control. Between herd moves encompassed 85 samples, eight of which were conjunctival swabs, whilst within herd moves and social stability management periods only contained trunk swabs (*n* = 55 for within, *n* = 25 for social stability).

### 2.5. Statistical Analysis

The following statistics were performed in R (v 4.0.3) and results were deemed significant at an α-value of *p* < 0.1 due to the sample size. EF–1–α Cts and normalised EEHV shedding (ΔCt) distributions were evaluated for normality using a histogram, normal Q-Q plot and Shapiro–Wilk test. Outliers were identified using box plots. Generalised estimating equations (GEE), designed specifically for longitudinal studies, were used to evaluate the effects of management periods on EEHV status (whether an elephant is negative or positive, i.e., exhibiting a Ct below 43) [50]. Exchangeable working correlation matrices accounted for repeated measures within the elephant and sample types were treated equally. Results were presented as odds ratios (OR) with 95% confidence intervals (CI). Sample type and elephant ID served as predictors for EEHV status in a GEE before assessing the odds of EEHV shedding associated with various management periods. If this GEE indicated sample type and elephant ID did not influence EEHV status, the elephant characteristics, management diary notes, and DNA extraction absorption profiles were cross-referenced to speculate on the aetiology of any insignificant shedding anomalies.

A GEE for EEHV status (using a binomial family distribution) determined whether social change influenced the odds of testing positive for EEHV relative to the control period in October 2020. A GEE (using a Gaussian family distribution) identified any differences in EEHV shedding (i.e., normalised shedding, ΔCt) during periods of social change compared to social stability. Results were presented as estimates with associated standard errors, and residuals were evaluated for normality to confirm model assumptions. To isolate the individual effects of between and within herd moves, these two GEEs were repeated, and a post hoc Tukey test revealed which posed the strongest influence on EEHV status.

To evaluate whether recrudescent cases exhibit a similar delay to those of primary infections [31,51] in the commencement of shedding following social change, samples collected within a week of the first event were excluded (i.e., 30 October 2019, 2 November 2019, 2 November 2020, 6 November 2020). A GEE analysing the effect on EEHV status was repeated, where any relative increase in the odds of testing positive was considered attributable to a one-week delay in shedding onset.

## 3. Results

### 3.1. DNA Extraction and qPCR Normalisation

DNA extractions were successful for all samples (*n* = 165) and all extractions yielded sufficient DNA for qPCR, as determined by the spectrophotometer. Housekeeping gene EF–1–α was identified in all extractions, indicating samples contained elephant DNA and lacked PCR inhibitors (Figure 2). The average EF–1–α and EEHV 1a/b PTC Cts were 31.32 and 28.22 respectively; samples deviating below or above this threshold were controlled for via the use of Pfaffl’s (2001) ΔCt normalisation method. Although EF–1–α Cts ranged considerably (range Cts = 24.25–42.46) and the data failed the Shapiro–Wilk test (*p* < 0.001), the histogram and Q–Q plot appeared normal (Figure A1). There were no trends suggesting shipping jeopardised sample integrity. Six EF–1–α Cts were outliers; however, five of these had undetectable levels of EEHV and, thus, did not influence GEE analyses. The remaining EF–1–α outlier (26.63 Cts) was detected from a sample provided by Cow 4 during between herd moves (2 November 2019), which identified EEHV 1a/b (37 Cts). All samples were included in subsequent analyses; normalisation eliminated all outliers and ΔCt data exhibited a normal distribution (Shapiro–Wilk *p* = 0.89) (Figure A2).

### 3.2. Shedding Events and Confounding Variables

Sixteen EEHV 1a/b shedding events were identified, but none was positive for EEHV 4a/b (Figure 3). No samples were identified to have more than one EEHV genotype. Neither elephant ID (Bull *p* = 0.70, Cow 1 *p* = 0.196, Cow 2 *p* = 0.72, Cow 3 *p* = 0.69, Cow 4 *p* = 0.69) nor sample type (conjunctival swabs *p* = 0.571, trunk swabs *p* = 0.313) significantly influenced the odds of testing positive for EEHV. Only one positive sample (collected from Cow 2) was detected throughout social stability, indicating no elephant was a continuous shedder and, thus, this period served as a suitable control. Two (Cow 4’s 2 November 2019; Bull’s 18 December 2019) and four (Cow 3′s 20 and 23 November 2020; Cow 4′s 13 and 16 November 2020) samples collected during between and within herd moves, respectively, did not coincide with other management events (Table 2).

### 3.3. The Influence of Social Change on the Odds of EEHV Recrudescence and Shedding Load for Individual Elephants

The period of social change demonstrated fifteen EEHV 1a/b shedding events, while the control period encompassed one. Social change increased the odds of testing positive for EEHV 1a/b relative to social stability; however, the effect was not significant (OR = 2.88, 95% CI = 0.363–22.8, *p* = 0.317). Compared to the singular positive sample collected during the control period, those obtained throughout social change exhibited significantly less viral DNA loads per sample (+3.119 ΔCts ± 0.483, *p* = 1.1 × 10^−10^). Residuals were normally distributed, confirming model fit (Shapiro–Wilk test *p* = 0.8) (Figure A3).

### 3.4. The Respective Effects of between and within Herd Moves on the Odds of EEHV Recrudescence and Shedding Load for Individual Elephants

Between and within herd moves encompassed five and ten EEHV 1a/b shedding events, respectively. Though the influence was not significant, between and within herd moves were associated with an increase in EEHV 1a/b detection odds compared to social stability (between herd moves OR = 1.5, 95% CI = 0.167–13.5, *p* = 0.717; within herd moves OR = 5.33, 95% CI = 0.644–44.2, *p* = 0.121). Between and within herd moves also exhibited significantly higher Cts, and, therefore, lower viral DNA loads per sample, relative to the singular positive control (between herd moves +3.44 ΔCts ± 1.28 *p* = 0.0074; within herd moves +2.95 ΔCts ± 0.311 *p* < 2 × 10^−16^). Residuals were normally distributed (Shapiro–Wilk test *p* = 0.8) (Figure A3). Between herd moves demonstrated the lowest mean normalised EEHV DNA load among all management periods and shedding was most common throughout within herd moves (18.18% of samples were positive) (Figure 4). This trend was reflected in the post hoc analysis, which revealed within herd moves imposed a significantly stronger influence on EEHV status compared to between herd moves (*p* = 0.0425).

### 3.5. Characteristics of Recrudescent Viral Shedding

Samples obtained within the first week of a social change period may have been reflective of previous, rather than current, management. When excluding the first week of each period, the odds of EEHV recrudescence increased for all social change management periods and revealed a significant influence from within herd moves (social change OR = 3.27, 95% CI = 0.412–26, *p* = 0.262; between herd moves OR = 1.6, 95% CI = 0.178–14.4, *p* = 0.675; within herd moves OR = 6.86, 95% CI = 0.823–57.1, *p* = 0.0751). The swiftest EEHV 1a/b shedding onset for both between and within herd moves occurred after eight days (Figure 5). Cow 2 appeared to shed most frequently, providing five positive samples throughout the study, while Cow 1 shed once. Most shedding events were singular; continuous shedding (meaning consecutive positive samples) occurred three times and only during within herd moves, the longest being a week from Cow 2. Two consecutive shedding events successively decreased viral loads with time, yet Cow 4 exhibited an inverse response and peaked viral shedding two weeks after the first within herd move.

## 4. Discussion

To the authors’ knowledge, this is the first longitudinal study examining the independent effects of between and within herd moves on EEHV recrudescence. All management periods were associated with increased odds of detectable EEHV 1a/b, however, only the effect of within herd moves was statistically significant (OR = 6.86, 95% CI = 0.823–57.1, *p* = 0.0751, post hoc *p* = 0.0425). Both management periods could potentially increase clinical HD risk via inverse effects: between herd moves elevate the chance of one-off exposure to a high viral load or a new virus genotype, whilst within herd moves could promote herd-wide EEHV recrudescence, increasing overall background but manageable viral load exposure. From an epidemiological perspective, these findings suggest that exposure to new elephants and novel mixing should be considered as EEHV exposure risks to susceptible elephants, if present.

There are several limitations associated with this study. As EEHV 1a and 1b are indistinguishable by this qPCR, co-infections may have been concealed or consecutive shedding events may have encompassed sequential infections [24,30,52]. Four of five sampled elephants had previously resided elsewhere; relocation can expose elephants to novel strains [12,13], increasing the odds of co-infection. A larger sample size would have been preferable to establish a credible control baseline or equivalent sample sizes among management periods. Time and resource constraints have also prevented technical qPCR replicates, limiting statistical accuracy. It is noteworthy to mention that detecting EEHV shedding onset can be influenced by the employed methods (i.e., efficiency of sampling, DNA extraction and qPCR). Unknown factors could have also promoted recrudescence; for example, a latent case in Napoli reactivated during a period of *Salmonella* septicaemia [53]. Taking these challenges into account, biological implications of these results remain valid.

The EF–1–α gene served as an internal control and host DNA was identified in all samples. For Asian elephants, although cotton swabs produce approximately three CTs higher (equivalent to 10-fold) nucleic acids than whole blood [46], because all swabs were identical, detection of host DNA proved reliable and comparable among types. The six EF–1–α Ct outliers may indicate some variability in sample collection; however, these did not influence inferential statistics since data normalisation was highly effective. Four of six EF–1–α Ct outliers arose from conjunctival swabs, though only one detected EEHV 1a/b (Cow 4′s 2 November 2019 sample); this was as expected as conjunctiva has the least cellular turnover. To ensure consistent sampling among elephants and to increase sample size, both sample types were included. However, previous research using a nearly identical DNA extraction protocol found conjunctival sampling to be more sensitive for EEHV 1a/b [6]. This relationship was reflected for EF–1–α, as Figure 2 suggests greater sensitivity from conjunctival swabs. Nevertheless, inferential findings remain legitimate as the sample type GEE found no significant effect on EEHV detection odds [49]. Initial validation of this real time PCR was carried out by Stanton et al. (2010), in which they tested the PCR against all their EEHV-1 positive archival samples, including a case involving an EEHV1a/b chimeric genome and the test provided a positive result for all [30]. We have also compared complete DNA sequences for the Major DNA-Binding Protein (MDBP) gene, the gene to which the EEHV-1 primers and probe are designed, from those EEHV-1a and 1b sequences available in the GenBank and from unpublished APHA archive. The analysis indicated a nucleotide identity of 99.2–100% for this gene, a relatively stable gene considering its vital role in virus replication. The sequences of the EEHV-1 primers and probe, however, were identical to those of the sequences analysed. Nonetheless, the potential of EEHV-1 false negatives could not be overlooked, but their likelihood is negligeable.

EEHV 1a/b shedding appeared to commence six days after a stressful event—one day less than previously described primary cases [31]. Three samples collected less than six days after a confounding management event detected EEHV 1a/b; however, recrudescence was likely initiated by other variables. Cow 1 started shedding EEHV 1a/b two days after barrier-protected tactile touch with the bull (on 18 November 2019), but aggression from Cow 2 thirteen days prior, or vaccination training and injection six days prior, may also have contributed to virus reactivation and shedding [54]. The bull shed EEHV 1a/b three and four days after mating events with Cows 3 and 1, respectively. Other mating events took place beforehand, which could have equally promoted virus reactivation. It is possible that the act of mating and its associated behaviours (e.g., smelling of vulvas and urine) physically distribute EEHV among the herd [55]. Cow 3 (39 years old) shed EEHV 1a/b seven days after mating (on 27 November 2020), yet the Bull almost exclusively mixed with Cow 1 (eight mating events), who did not recrudesce during this time. Cow 3′s oestrus cycles are flatlining, so she may have been unreceptive to mating, and the resulting stress promoted EEHV 1a/b reactivation. If future research suggests EEHV is only infectious at high viral loads, appropriate management for within herd moves could expose susceptible individuals to minimal/moderate viral loads and provide an opportunity for immunity development. Captive Asian elephant cows typically ovulate and are receptive to mating for two to ten days [56]. After a cow’s receptive period, a bull’s presence can be socially challenging and induce immunosuppression. A cost-effective and efficient suggestion to detecting a cow’s follicular phase through urinary pheromones would involve presenting consistently collected cow urine samples to the bull [57]. If a bull indicates a cow is ovulating, it may prove advantageous to conduct within herd moves for one week. This could simulate in-situ mixing and allow adequate time for elephants to mate, possibly reactivate shedding of minimal viral loads, meanwhile limiting prolonged exposure.

Veterinary procedures (blood draws and ultrasounds), training (operant and desensitisation), and foot care occurred frequently during non-shedding periods and, hence did not appear to be associated with EEHV recrudescence. Previous research did not identify any associations between EEHV detection and behaviour, environment, management (e.g., sleeping activities and staff changes), or medical changes [39].

In this study, some events noted within management diaries coincided with and may have promoted reactivation. The only positive control sample, which was Cow 2′s highest viral load, may have been promoted by a split toenail occurring nine days prior. Considering the GEEs comparing viral loads were relative to this single positive control sample, it would be inappropriate to conclude that social change promotes relatively lesser viral loads. Nine days prior to one of Cow 2′s other shedding events (27 November 2020), conspecific aggression ensued. Additionally, Cow 2′s shedding may have been influenced by her pregnancy [43]. A shedding event also occurred fifteen days after the initiation of the Bull’s musth. Cow 1’s tetanus vaccination may have also triggered immunosuppression and subsequently EEHV 1a/b recrudescence. In other mammals, effective vaccines are known to be associated with a temporary period of immunosuppression [54]. Not only was this the only vaccination administered during the study period, but it also occurred exactly six days prior to Cow 1′s only shedding event (20 November 2019). If this study were to exclude all samples potentially associated with confounding variables, the effect of social change on EEHV recrudescence remains apparent: two and four positive samples would remain exclusively attributable to between and within herd moves, respectively.

These findings support the previous recrudescence trends documented by Atkins et al. (2013) and Sanchez et al. (2016), though are opposite to those in Hengtrakul et al. (2020) [39,41,42]. Both management actions incited EEHV reactivation in 80% of elephants in this study, so it is possible that external stress caused recrudescence for the entire Atkins et al. (2013) herd [41]. Inverse shedding responses occurred during within herd moves, similarly to the effect of between herd moves as described by Sanchez et al. (2016) [42]. For most events (75%) EEHV shedding followed a decline over time. Only Cow 4 exhibited an increase in viral shedding; as the dominant female she may have been socially challenged by the Bull’s presence. Hengtrakul et al. (2020) found that recrudescent EEHV in young elephants reactivated more often [39], yet in this study, the virus in the youngest cow (Cow 1, 11 years old) reactivated only once (20 November 2019), six days after her tetanus vaccination. For all other sampled elephants, the virus recrudesced on a minimum of two separate occasions. This study reveals shedding rates can vary per individual, and do not appear directly associated with age but rather novel mixing.

The aetiology of the high level of EEHV-HD mortality rates in captivity remains unknown. Some management actions, such as between herd moves, are scenarios specific to captivity. In-situ, social fluctuation and environmental olfactory cues among herds may promote constant shedding of low viral loads and therefore ample circumstances for susceptible individuals to build immunity. Management-derived social changes ex-situ may implement interactions otherwise not observed in the wild, limiting opportunity for immune system development thus elevating HD mortality risk. Haycock (2020) suggested increased frequency of fatal HD in captivity may be attributable to smaller herd sizes, since vulnerable calves are not as intermittently exposed to EEHV [26]. Herd size can affect numerous factors, including mating frequency and social interactions. The study herd is relatively small compared to free-living conspecifics, which can include over 100 individuals [58]. Research investigating the epidemiological kinetics of a larger group may reveal the key to developing immunity, because elevated intraspecies interactions (e.g., birth events, mixing of elephants, mating events) give rise to more opportunities for recrudescence and low-level viral exposure [1].

The findings of this study reveal several implications to captive elephant management. Management-derived social changes should be considered as EEHV exposure risks to susceptible Asian elephants. The importance of EEHV monitoring is clear and between herd movements should be avoided if a naïve individual is present within the receiving herd. Future research should focus on longitudinal, consistent sampling of multiple institutions with various herd sizes, alongside disease and external stress monitoring. Comparison of facilities with and without a history of fatal HD could also illuminate the non-viral and non-host factors attributable to EEHV mortality. It is evident, however, that between and within herd moves have epidemiological consequences, and if EEHV is to be managed appropriately, it is crucial to recognise their associated risks and proactively monitor susceptible individuals.

## 5. Conclusions

Understanding how to mitigate EEHV infections is not only a conservation, health, and welfare issue, but also a moral obligation considering the frequency of captive HD mortalities and the influence of management, as newly described by this study. Stem cell and pathophysiology research are taking place [59,60,61,62], as is experimental vaccine testing. Nevertheless, it can take years to adopt a successful captive vaccination programme and, thus, ascertaining which management actions elevate risk is necessary. Management-derived social change causes EEHV reactivation in Asian elephants. EEHV recrudescent elephants appear to commence shedding anywhere from six days to (at least) two months following between and within herd moves, which instigates close monitoring of at-risk Asian elephant calves after such events.

## Figures and Tables

**Figure 1 viruses-14-00229-f001:**
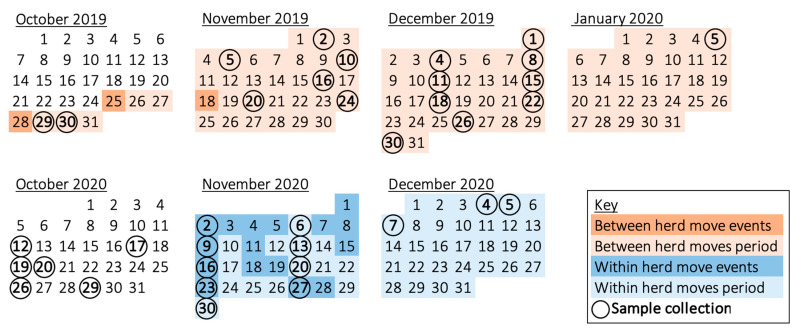
Calendar of the two virus screening periods, illustrating the dates of sample collection and management events. Circled dates describe sample collection from five elephants—except, in the case of consecutive dates (19–20 October 2020 and 4–5 December 2020), at which times the four cows were swabbed the first day and the bull was swabbed the second. Two bulls were moved away on 25 October 2019, a new bull moved into his respective enclosure on 28 October 2019, and tactile contact through a barrier was allowed with the cows after 18 November 2019; hence, samples collected from 30 October 2019 to 5 January 2020 were assigned as the ‘between herd moves’ period. Direct mingling for breeding took place throughout November 2020; thus, samples from 2 November 2020 to 7 December 2020 were designated as the ‘within herd moves’ period. The control period of social stability occurred throughout October 2020. ‘Social change’ represents the between and within herd moves management periods.

**Figure 2 viruses-14-00229-f002:**
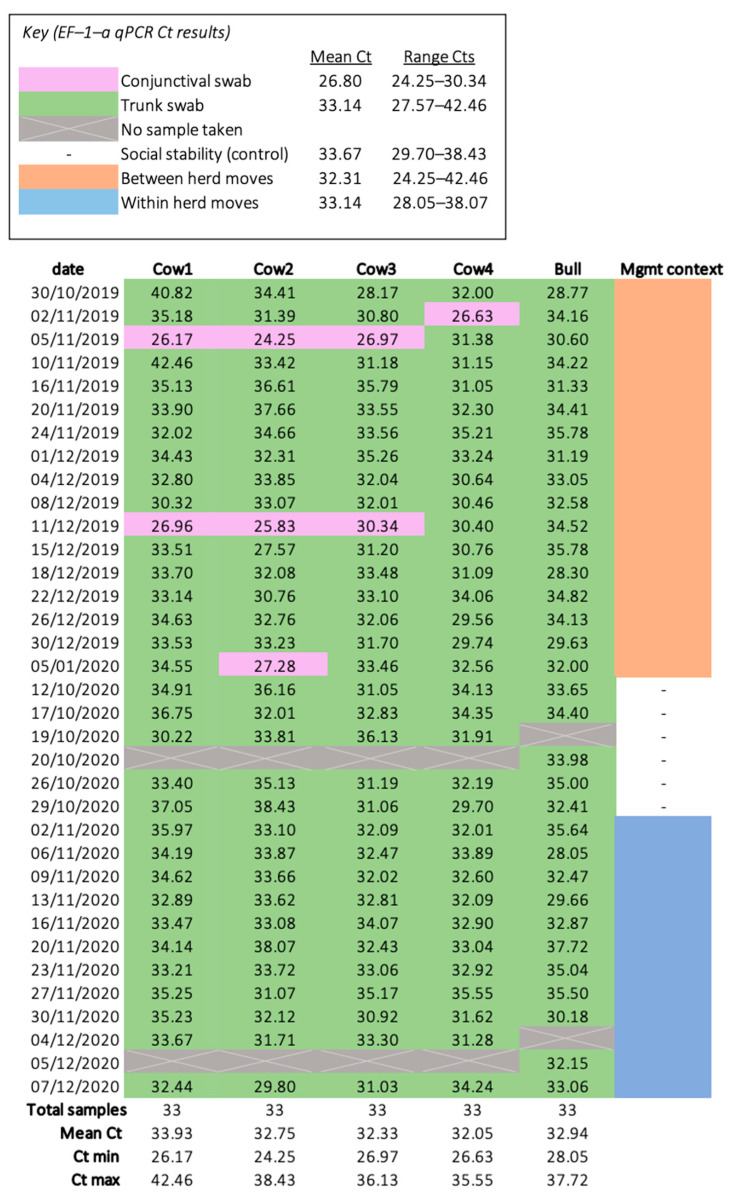
Heatmap of the housekeeping gene (EF–1–α) qPCR results. All samples had a detectable level of EF–1–α and, therefore, no samples were excluded from statistical analyses.

**Figure 3 viruses-14-00229-f003:**
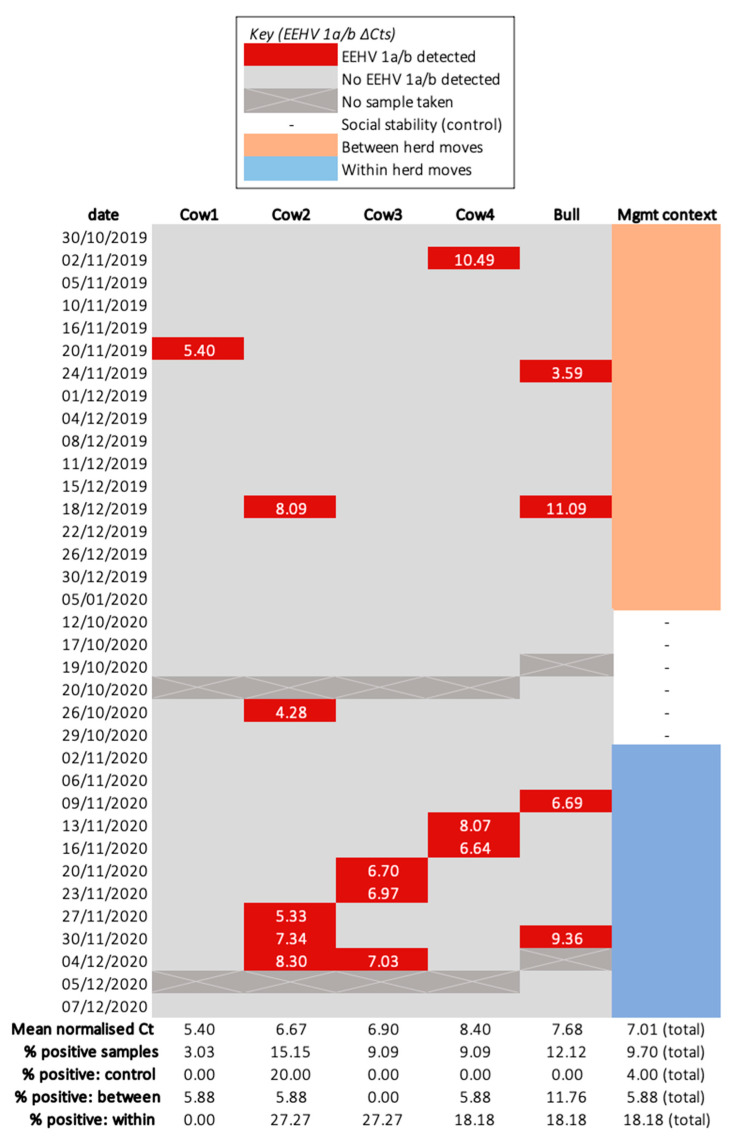
Heatmap of normalised EEHV 1a/b shedding (ΔCt) for each sampled elephant. The percentage of samples exhibiting an EEHV 1a/b and EF–1–α Ct below 43 is indicated within the “% positive” rows for each individual elephant and the total herd. This value was calculated by dividing the quantity of samples possessing EEHV 1a/b by the total number collected within the corresponding period and multiplying by 100.

**Figure 4 viruses-14-00229-f004:**
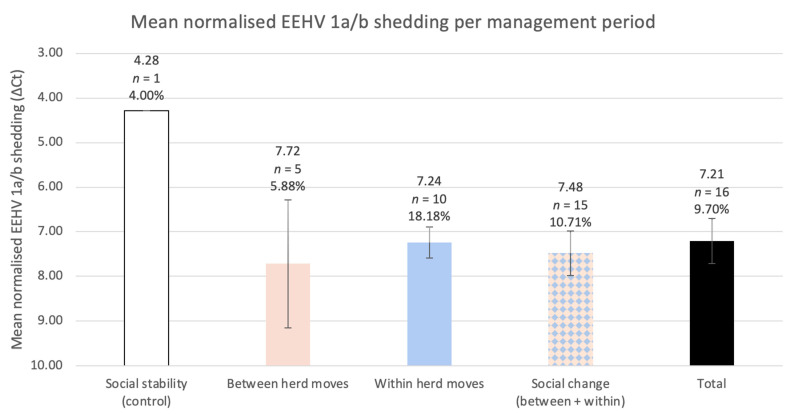
The study herd’s mean normalised EEHV 1a/b DNA load for each management period, with corresponding error bars (95% confidence intervals). The percentage of positive samples is listed below each sample size and was calculated identically to those in Figure 3. Low Cts represent higher viral DNA, therefore, the y-axis is inverted to visualise the relationship between ΔCt (normalised shedding) and the relative viral DNA. GEEs informed by normalised shedding revealed significantly less viral DNA per sample within all social change periods compared to the singular positive control sample (social change *p* = 1.1 × 10^−10^; between herd moves *p* = 0.0074; within herd moves *p* < 2 × 10^−16^).

**Figure 5 viruses-14-00229-f005:**
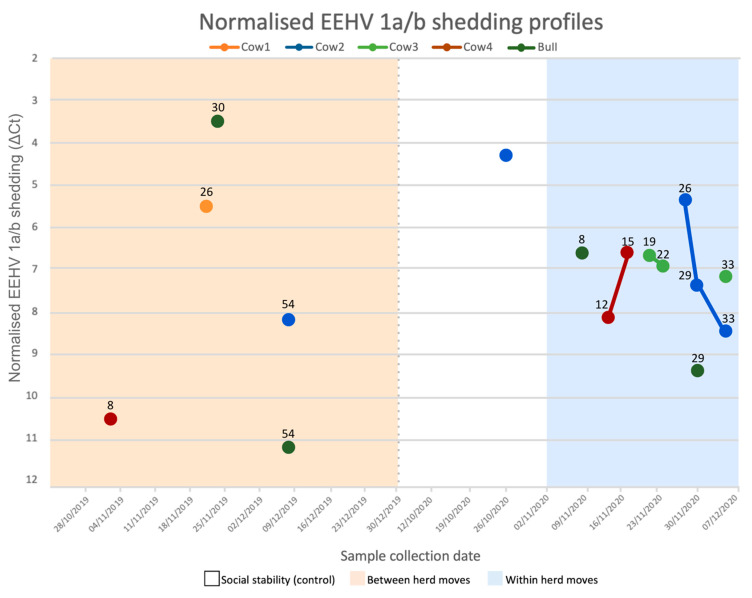
EEHV 1a/b normalised shedding profiles for each sampled Asian elephant. Values above each data point indicate the number of days post the first management event (i.e., 25 November 2019 for between herd moves and 1 November 2020 for within herd moves). Samples that are joined by a line were collected over sequential sampling dates. The y-axis is flipped to appropriately represent the inverse relationship between ΔCt (normalised shedding) and relative viral DNA loads, as low ΔCts represent high viral DNA. Sample collection dates are listed by week, beginning on Mondays. The dotted grey line indicates a 10-month separation between periods. Tactile touch behind a barrier was allowed following 18 November 2019. This shedding profile describes positive EEHV samples only. Elephants did not present detectable EEHV 1a/b nor 4a/b DNA at all other sampling points.

**Table 1 viruses-14-00229-t001:** Primers, hydrolysis probes and PTCs employed to detect EEHV 4a/b [46], and EF–1–α. EEHV 1a/b primers and probes were from Stanton et al. (2010) [30].

	Forward primer	5′-GAT CCA CAA GGA GTT CGG-3′
	Reverse primer	5′-GTC CGT GAT ATT TAC GTK ACT-3′
EEHV 4a/b	Hydrolysis probe	5′-[FAM] AAT AGT CGC CAC GTC TCC ATG [MGBEQ]-3′
	PTC	5′-CGC AGG TGC GCA CGA TCC ACA AGG AGT TCG GGT TTA ATA GTC GCC ACG TCT CCA TGC TGC TGC GGG ACA GTC ACG TAA ATA TCA CGG ACG TGG CCA GCA A-3′
	Forward primer	5′-CCA CAT CAA CAT CGT CGT C-3′
EF–1–α	Reverse primer	5′-TTC CCA TCT CAG CAG CTT C-3′
	Hydrolysis probe	5′-[FAM] AGT CCA CCA CTA CTG GTC ACC TGA TCT ACA A [BHQ1]-3′

**Table 2 viruses-14-00229-t002:** A list of notable management events occurring throughout the study period which may have influenced EEHV recrudescence. Data was extracted from keeper management diaries to identify a broad correlation with other management events coinciding with shedding. Weekly foot care and training (operant and desensitisation) occurred for each elephant throughout every management period. Veterinary procedures, such as blood draws and injections, also frequently took place during non-excretory periods. Days until the next shedding are presented in order of elephant ID, where “-” indicates no EEHV was detected after the management event within the appropriate period.

Management Period	Date	Elephant ID	Management Event	Next Shedding (Days)
	31 October 2019	Bull	Desensitisation training to steam train	24
	4 November 2019	Cow 2	Ultrasound scan—parturition onset	44
	7 November 2019	Cow 1	Small wound from Cow 2’s tusks	13
	7 November 2019	Cow 2	Ultrasound scan—suspected parturition	41
Between herd moves	13 November 2019	Cow 1	Mock vaccination	7
	13 November 2019	Cow 2	Ultrasound scan—foetal death suspected	35
	14 November 2019	Cow 1	Tetanus vaccination	6
	18 November 2019	Cow 1; Bull	First allowed tactile touch between Bull and cows; all cows interacted though attention was focused on Cow 1	2; 6
	25 November 2019	Cow 2	Ultrasound	23
Social stability	17 October 2020	Cow 2	Split toenail	9
	1 November 2020	Cow 1 + Bull	Mating event	-; 8
	2 November 2020	Cow 1 + Bull	Mating event	-; 7
	4 November 2020	Cow 1 + Bull	Mating event (unsuccessful, Cow 1 not interested)	-; 5
	5 November 2020	Cow 1 + Bull	Mating event (unsuccessful × 3, Cow 1 not interested)	-; 4
	11 November 2020	Cow 1 + Bull	Mating event	-; 19
Within herd moves	14 November 2020	Cow 1 + Bull	Mating event	-; 16
	15 November 2020	Cow 1 + Bull	Mating event (unsuccessful, Cow 1 not interested)	-; 15
	15 November 2020	Bull	Musth initiated	15
	18 November 2020	Cow 2	Bullying Cow 4	9
	23 November 2020	Cow 1 + Bull	Mating event (unsuccessful × 2, Cow 1 not interested)	-; 7
	27 November 2020	Cow 3 + Bull	Mating event	7; 3

## Data Availability

The data presented in this study are available on request from the corresponding author, stitus20@rvc.ac.uk.

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
