# Peer review of "Effects of between and within Herd Moves on Elephant Endotheliotropic Herpesvirus (EEHV) Recrudescence and Shedding in Captive Asian Elephants (Elephas maximus)"

_viruses, 2022, doi:10.3390/v14020229_

Round 1

Reviewer 1 Report

This is a new manuscript from Titus et al "Effects of between and within herd moves on elephant endothelial herpesvirus (EEHV) recrudescence and shedding in captive Asian elephants". The authors have collected multiple primary samples from 5 elephants at one zoo over several months. The time frame includes potentially stressful experiences that could affect the emergency from latency of EEHV. They employ quantitative PCR of a host gene and EEHV to measure potential viral shedding. They conclude that the introduction of a bull to the herd led to increased shedding. This leads to implications for herd management to minimize haemorrhagic disease and death.

EEHV has a significant impact on the conservation of Asian elephants. The study does have limitations such as the limited number of elephants evaluated. However, it is understandably difficult to have a large sample number for this type of study. The authors acknowledge this in the discussion. Regardless of this, the study is very suggestive of herd management practices as this species nears extinction.

There are some issues that could be addressed by revision of the text.

Most readers will not be particulary familiar with EEHV. It would be very helpful to include a brief discussion of the transmission routes particularly to differentiate between the two types of contact experienced by the herd. Please also include the cell/tissue tropism of the virus.

While DNA viruses are not as mutagenic as RNA viruses, there is still some genetic variability. Some discussion of the mutagenic rate and/or confidence in the genetic stability of the primer binding sites should be included. What is the probability that some EEHV will be missed if these primer sequences were mutant in the animal?

It looks like the first data listed in Table 2 has a typo "31/20/19".

Author Response

Point 1: Most readers will not be particularly familiar with EEHV. It would be very helpful to include a brief discussion of the transmission routes particularly to differentiate between the two types of contact experienced by the herd. Please also include the cell/tissue tropism of the virus.

Response 1: Thank you for this feedback and the author team agree that more background information on EEHV would be advantageous. In the second paragraph of the introduction, more detail was added regarding incubation periods, variaemia and shedding timelines. A description of probable transmission routes was mentioned and how these are related to the social change periods researched in this article (between and within herd moves). Additionally, cell and tissue tropism for the relevant genotypes (EEHV 1a and 4) are included.

Point 2: While DNA viruses are not as mutagenic as RNA viruses, there is still some genetic variability. Some discussion of the mutagenic rate and/or confidence in the genetic stability of the primer binding sites should be included. What is the probability that some EEHV will be missed if these primer sequences were mutant in the animal?

Response 2: The original research describing this assay (Stanton et al. 2010) validated procedures against archived samples. We have also compared complete DNA sequences for the Major DNA-Binding Protein (MDBP) gene, the gene to which the EEHV-1 primers and probe are designed, from those EEHV-1a and 1b sequences available in the GenBank and from unpublished APHA archive. In the third paragraph of the discussion, we elaborated on Stanton et al.’s findings and our GenBank/APHA archive search. The analysis indicated a nucleotide identity of 99.2%-100% for this gene; therefore, the likelihood of false negatives are negligible.

Point 3: It looks like the first data listed in Table 2 has a typo "31/20/19".

Response 3: Yes, thank you for catching this. The date was fixed to 31/10/19 in Table 2.

Reviewer 2 Report

The authors invstigated the effect of captive herd management on elephant endotheliotropic herpesvirus (EEHV) shedding. EEHV infection is the leading cause of death in captive Asian elephant calves. The manuscript is scientifically written. Experiments were well designed to achieve the purposes of this study. However, the manuscript should be corrected as indicated below.

L42: Reference 2 is not a scientific paper. Another appropriate paper should be cited.

L86-87: The authors wrote "Some facilities interestingly have not experienced ..." without citation. How did the authors know these data. The authors should show the scientific source of the description.

L86-92: Description in this paragraph lacks any scientific evidences. The authors should cite appropriate references for each description.

L265: The authors describe "Only one positive sample was detected". Which is the only one sample, Cow2 or another? The authors should show which is the only one sample.

L304: The authors describe "(18.18% of samples were positive)". The percentage cannot be found in Figure 4. Where does the percentage come from? An explanation of the percentage should be described.

L363-364: The authors describe "EF-alpha was chosen...". However, the authors detected host DNA but not cDNA derived from host mRNA. Therefore any target can be used for host DNA detection. It is no need to emphasize EF-1 alpha here. The authors should discuss on detection of host DNA.

L413: The authors describe "Cow 1's tetanus vaccination may have also triggered immunosuppression ...". The authors should show scientific data supporting why the vaccination might have triggered immunosuppression. 

L423 and 430: The authors cited a reference by Sanchez et al. (2016). However, the reference is an abstract and does not show any information in detail. The authors should cite another appropriate article.

L438: The authors described "Haycock (2020) suggested ...". There is no reference number. Which is a reference by Haycock?

L443-444: It is unclear why research investigating the epidemiological kinetics of a large group may reveal the key to developing resistance. The authors should add explanation supporting this description.

L559-560: Reference 12 might be "Viral gene subtyping of eighteen North American cases of EEHV hemorhagic disease. Proceedings of the International Elephant Conservation and Resarch Symposium. Bangkok, Thailand; 2009.

L562-563: Reference 13 might be "Pathogenesis and ... betaherpesvirinae. Proceedings of the Proceedings of the International Elephant Conservation and Resarch Symposium. Florida; 2008.

L626-629: This is an abstract in 2016 Joint AAZV/EAZWV/IZW Conference Proceedings. The authors should describe the correct source.

L637-639: The authors should show URL of the reference 40.

L676-677: The title of the reference 56 is wrong. It should be "Establishing assays to evaluate T cell responses to Elephant Endotheliotropic Herpesvirus (EEHV) ".

L678-679: The reference 57 is not cited in the manuscript.

Author Response

Point 1: L42: Reference 2 is not a scientific paper. Another appropriate paper should be cited.

Response 1: This source was replaced with a more appropriate reference by Howard and Schaftenaar (2019).

Point 2: L86-87: The authors wrote "Some facilities interestingly have not experienced ..." without citation. How did the authors know these data. The authors should show the scientific source of the description.

Response 2: This paragraph discusses Perrin et al.’s (2021) paper, who anaylsed the risk of EEHV fatality associated with history of EEHV-HD death in facilities across Europe. Although these institutions are not explicitly stated in the published version of Perrin et al.’s article, we have seen the raw data in which her article discusses and are aware of the locations without a history of EEHV-HD deaths through personal experience and anecdotal evidence. Personal communications from Dr Perrin is now cited in this sentence to support our claim.

Point 3: L86-92: Description in this paragraph lacks any scientific evidences. The authors should cite appropriate references for each description.

Response 3: Similarly to response 2, this paragraph was exploring Perrin et al.’s (2021) article, however the authors do acknowledge more citations would have clarified the intended message. Appropriate references were added and sentence structure was fixed so as to remove any ambiguity regarding which hypotheses have already been published by Perrin et al..

Point 4: L265: The authors describe "Only one positive sample was detected". Which is the only one sample, Cow2 or another? The authors should show which is the only one sample.

Response 4: Yes, you are correct that this sentence was referencing the one positive sample collected from Cow 2 during the period of social stability. This clarification was added to the text.

Point 5: L304: The authors describe "(18.18% of samples were positive)". The percentage cannot be found in Figure 4. Where does the percentage come from? An explanation of the percentage should be described.

Response 5: These percentages are listed in Figure 3. However, we do recognise they were stated in a sentence referencing Figure 4, so thank you for pointing that out. Percentage values were added to Figure 4, as well as a reference to these in the figure caption and a more detailed description of how we calculated these values in the caption for Figure 3.

Point 6: L363-364: The authors describe "EF-alpha was chosen...". However, the authors detected host DNA but not cDNA derived from host mRNA. Therefore, any target can be used for host DNA detection. It is no need to emphasize EF-1 alpha here. The authors should discuss on detection of host DNA.

Response 6: This is a valid point – thank you for mentioning this. The first sentence in the discussion’s third paragraph has been edited to omit our reasons why EF1-a was chosen as an internal control. We retained the point that EF1-a was identified in all samples, considering this naturally comments on the detection of host DNA. A further mention of host DNA detection with respect to swabs and the possibility of false negatives was also included.

Point 7: L413: The authors describe "Cow 1's tetanus vaccination may have also triggered immunosuppression ...". The authors should show scientific data supporting why the vaccination might have triggered immunosuppression.

Response 7: Considering this is one event, we lack data trends and therefore can only discuss the apparent correlation between the only instance of EEHV 1a/b recrudescence in Cow 1 and the timing of her vaccination. Further detail and a reference discussing post-vaccination immunosuppression in canines were added to this paragraph.

Point 8: L423 and 430: The authors cited a reference by Sanchez et al. (2016). However, the reference is an abstract and does not show any information in detail. The authors should cite another appropriate article.

Response 8: The authors recognise this reference is an abstract from the 2016 Joint AAZV/EAZQ/IZW Conference Proceedings, however we would argue the case presented is described in enough detail to discuss the inverse shedding responses observed during between herd moves. This is the only published literature currently available describing a case where between herd moves may have been associated with EEHV reactivation. Because another comparable article does not yet exist, the authors would be keen to acknowledge Sanchez et al.’s (2016) case in the perspective of our findings.

Point 9: L438: The authors described "Haycock (2020) suggested ...". There is no reference number. Which is a reference by Haycock?

Response 9: This was reference 26, which was added to the end of the sentence in brackets.

Point 10: L443-444: It is unclear why research investigating the epidemiological kinetics of a large group may reveal the key to developing resistance. The authors should add explanation supporting this description.

Response 10: Perrin et al. (2021) hypothesised that exposure to new elephants and calving events may influence EEHV reactivation and shedding. In larger herds, these events will occur more frequently solely due to the larger number of elephants present. The specific instraspecies interactions that would occur more frequently in a larger group and may be associated with EEHV recrudesence were added to this sentence to provide clarification.

Point 11: L559-560: Reference 12 might be "Viral gene subtyping of eighteen North American cases of EEHV hemorhagic disease. Proceedings of the International Elephant Conservation and Resarch Symposium. Bangkok, Thailand; 2009.

Response 11: We opted to use the free reference manager Zotero, which despite being double-checked, obviously came with its quirks. Thank you for checking our references and noticing this. The change has been updated.

Point 12: L562-563: Reference 13 might be "Pathogenesis and ... betaherpesvirinae. Proceedings of the Proceedings of the International Elephant Conservation and Resarch Symposium. Florida; 2008.

Response 12: Reference 13 was updated as suggested. The country where this symposium took place was also added to the citation.

Point 13: L626-629: This is an abstract in 2016 Joint AAZV/EAZWV/IZW Conference Proceedings. The authors should describe the correct source.

Response 13: Please see our response 8 detailing why the authors would like to retain discussion of Sanchez et al.’s (2016) abstract.

Point 14: L637-639: The authors should show URL of the reference 40.

Response 14: Doi’s were added to all published articles who were missing them and URLs were added to all PhD dissertations, including reference 40.

Point 15: L676-677: The title of the reference 56 is wrong. It should be "Establishing assays to evaluate T cell responses to Elephant Endotheliotropic Herpesvirus (EEHV) ".

Response 15: Again, thank you for checking our references and noticing this. The change has been updated.

Point 16: L678-679: The reference 57 is not cited in the manuscript.

Response 16: Thank you for catching this; unfortunately, it was another Zotero error. This reference was supposed to serve as a citation for section 2.2., and is now updated.